# Probability-boosting technique for combinatorial optimization

Sanpawat Kantabutra

Department of Computer Engineering, Faculty of Engineering, Chiang Mai University, Chiang Mai, Chang Wat Chiang Mai, Thailand

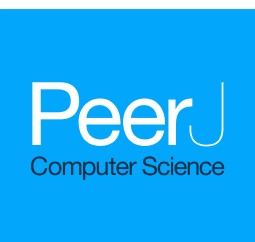

## ABSTRACT

In many combinatorial optimization problems we want a particular set of k out of n items with some certain properties (or constraints). These properties may involve the k items. In the worst case a deterministic algorithm must scan n−k items in the set to verify the k items. If we pick a set of k items randomly and verify the properties, it will take about $(n/k)^k$ verifications, which can be a really large number for some values of k and n. In this article we introduce a significantly faster randomized strategy with very high probability to pick the set of such k items by amplifying the probability of obtaining a target set of k items and show how this probability boosting technique can be applied to solve three different combinatorial optimization problems efficiently. In all three applications algorithms that use the probability boosting technique show superiority over their deterministic counterparts.

## INTRODUCTION

Combinatorial optimization is the study of finding an optimal item or an optimal set of items from a finite set of items, where the set of feasible solutions is discrete. Some of these combinatorial optimization problems are known to be hard while others are not. Often-mentioned problems in combinatorial optimization are the travelling salesperson problem, the minimum spanning tree problem, and the knapsack problem (*Cormen et al., 2001*). Solutions to combinatorial optimization problems vary from deterministic polynomial-time algorithms, to approximation algorithms, to linear programming, to randomized algorithms. We are interested in a type of combinatorial optimization that chooses $k$ out of $n$ items such that these $k$ objects meet some certain criteria. The eight queens puzzle, for example, can be viewed as choosing eight out of 64 positions on the $8 \times 8$ chessboard so that no two queens threaten each other (*Ball, 1960*). In social media networks some users are influential and have a greater power to relay information (*Qiang, Pasiliao & Zheng, 2023*). The problem of locating these $k$ out of $n$ users who can influence the most users within a limited time frame can also be viewed as the type of choosing $k$ out of $n$ items. Additionally, this particular type of problems appears in many *NP*-hard decision problems such as vertex cover problem, clique problem, dominating set problem, independent set problem, and set cover problem (*Garey & Johnson, 1990*) and these *NP*-hard problems have a wide range of applications.

Corresponding author
Sanpawat Kantabutra,
sanpawat@alumni.tufts.edu

In the worst case a deterministic algorithm must scan $n - k$ items in the set to verify the $k$ items. If we choose at random such $k$ items out of a total of $n$ items, it takes about $\binom{n}{k} \geq \left(\frac{n}{k}\right)^k$ verifications, which can be a really large number for some values of $n$ and $k$. More importantly, in some applications, the generation of all possible such $\binom{n}{k}$ combinations could take excessive amount of memory when the values of $n$ and $k$ are large. There exists an algorithm to generate each of the $\binom{n}{k}$ combinations uniformly at random (*Fan, Muller & Rezucha, 1962*; *Jones, 1962*; *Knuth, 1998*). In addition, there is also a family of randomized algorithms for choosing a simple random sample, without replacement, of $k$ items from a population of unknown size $n$ or of a known size that is too large to fit into main memory in a single pass over the items (*Vitter, 1985*). However, in either case, the number of verifications in the worst case is still large.

In the last two decades or so, computer scientists have witnessed a tremendous growth in the use of probability to solve difficult problems. The prevalent use of randomized algorithms is one of such examples. Randomized algorithms are algorithms that make random choices during their execution and then execute according to these random outcomes (*Mitzenmacher & Upfal, 2005*). For example, the protocol implemented in an Ethernet card uses random numbers to decide the next attempts to access the shared Ethernet communication medium when there is a collision of requests (*Hussain, Kapoor & Heidemann, 2004*). In this case the randomness is useful for breaking symmetry, preventing different cards from repeatedly accessing the medium at the same time. Other applications of randomized algorithms include Monte Carlo simulations (*Raychaudhuri, 2008*) and primality testing in cryptography (*Goldwasser & Kilian, 1999*). In these and many other important applications, randomized algorithms are significantly more efficient than the best known deterministic solutions. Moreover, in most cases the randomized algorithms are also simpler and easier to program.

In our case, we would also like to take advantage of the randomized approach to find the $k$ out of $n$ items. If we apply a straight forward uniformly randomized method to find such $k$ out of $n$ items, we will have a very small chance of success of only one in $\binom{n}{k}$. In this article we, therefore, consider a method to increase the probability of choosing $k$ out of $n$ items and illustrate the usefulness of our method in three different real-world applications. Note that if values of $n$ and $k$ are close, to find such $k$ items becomes easy. It is difficult to find such $k$ items when the values of $n$ and $k$ are far apart and we will only consider this latter case in this article.

Increasing a probability of some particular event of interest is not new. Obviously, appropriately reducing the size of a sample space can have such an effect. Given a set of $n$ numbers, the classic Insertion Sort algorithm (*Cormen et al., 2001*) reduces incrementally the space of all possible $n!$ permutations until it arrives at the sorted sequence of $n$ numbers. In machine learning the gradient descent algorithm (*Sra, Nowozin & Wright, 2011*) also reduces the sample space of all possible parameters at each iteration until it arrives at a local minima. There are also some other methods to increase the probability of the event of interest in literature. Consider the Solovay-Strassen primality test which always answers true for prime numbers and false with probability at least $\frac{1}{2}$ and true with

**Table 1  Advantages of the probability boosting technique.**

| Applications | With probability boosting | Deterministic methods |
|---|---|---|
| Online content optimization | Meet the frequency requirements for each rate while having a random display sequence of advertisements. | None exists. |
| Server selection in a heterogeneous environment | Obtain $k$ supercomputers with a very high probability and, therefore, a high speedup for processing as well as save a significant number of verification tests. | Need $n - k$ verification tests in the worst case to obtain $k$ supercomputers. |
| String comparison | Identify the $2k'$ differences with an average of $2(k' + i_1) < n - k$ positional checks with a good deviation from the average. | Require $n - k$ positional checks in the worst case to identify the $k$ differences. |

probability less than $\frac{1}{2}$ if the numbers are composite (*Solovay & Strassen, 1977*). To amplify the success probability, we can run this algorithm several times. If it returns true, then the number is prime. If it consecutively returns false responses $k$ times, the number is composite with probability at least $1 - 2^{-k}$. In general, any Monte Carlo algorithms with one-sided errors can use this probability amplifying technique. For two-sided error Monte Carlo algorithms, the probability can be boosted by running the algorithms $k$ times and using the majority of responses as an answer. In quantum computation probability or amplitude amplification is used in a family of quantum algorithms inspired by the Grover's search algorithm (*Brassard & Høyer, 1997*; *Brassard, HØyer & Tapp, 1998*; *Grover, 1998*). In addition, probability amplification is also used in two-way quantum finite automata (*Yakaryılmaz & Say, 2009*). To the best of our knowledge, no probability-boosting technique in combinatorial optimization similar to ours exists in literature.

Our contributions are as follows. We discuss the probability-boosting technique that can be used to increase the probability of choosing $k$ out of $n$ objects in the first section and then illustrate different uses of the technique in three applications. In the first application we use the probability boosting technique to help choose the set of $k$ online advertisements with a more expensive rate and a required frequency of display. In the second application we consider a heterogeneous environment of computer servers, in which there is a small set of $k$ powerful supercomputers. Our probability boosting technique is used to select this small set of $k$ supercomputers to effectively increase the overall processing speedup. We consider genetic comparison, in which two genetic strings are non-related, in the last application. We show that the probability boosting technique can successfully help to identify the $k$ differences in the genes. The superiority of our technique can be summed up in Table 1.

## THE PROBABILITY-BOOSTING TECHNIQUE

In this section we discuss a randomized technique that can be potentially used to solve many combinatorial optimization problems. Suppose we have $n$ different items and $k$ of these items are the items that we would like to obtain. If we simply choose $k$ items out of $n$ items uniformly at random, the probability is one in $\binom{n}{k}$ that we will obtain the desired items. This probability is small when $n$ is large and $k$ is small. In the rest of this article, unless stated otherwise, we assume that $1 \leq k < n$. Our technique is based on an

observation that if we choose $k + i$ items uniformly at random instead of just $k$, the probability that we will obtain a $(k + i)$-combination that includes the $k$ desired items will significantly increase. The following theorem describes this relationship. The proof of the theorem begins with an observation that when a $(k + i)$-combination of the set $S$ is picked uniformly at random, the probability of obtaining a targeted $k$-combination of the set $S$ equals $\frac{\binom{n-k}{i}}{\binom{n}{k+i}}$. This is because $k$ is fixed and the probability only depends on $n - k$ and $i$ for all such $(k + i)$-combinations of the set $S$. The proof then uses a lower bound to bound the probability of obtaining $\frac{1}{p}$, given $i$. Observe that in each step of the derivations of the inequalities the numerators get smaller while the denominator is fixed, making the inequalities hold.

**Theorem 1** (Probability Boosting). *Given constants $p$ and $i$, and any targeted $k$-combination of a set $S$ with size $n$, a $(k + i)$-combination of the set $S$ contains the targeted $k$-combination of the same set $S$ with probability at least $\frac{1}{p}$ if*

$$i \geq \left( \frac{n(n-1)(n-2)\dots(n-(k-2))(n-(k-1))}{p} \right)^{\frac{4}{3k+4}} - \frac{k}{4}.$$

*Proof.* Observe that when a $(k + i)$-combination of the set $S$ is picked uniformly at random, the probability of obtaining a targeted $k$-combination of the set $S$ equals $\frac{\binom{n-k}{i}}{\binom{n}{k+i}}$.

$$\frac{\binom{n-k}{i}}{\binom{n}{k+i}} = \frac{(k+i)(k+i-1)(k+i-2)\dots(k+i-(k-2))(k+i-(k-1))}{n(n-1)(n-2)\dots(n-(k-2))(n-(k-1))}$$

$$\geq \frac{(k+i)(k+i-1)(k+i-2)\dots(k+i-\frac{3k}{4})}{n(n-1)(n-2)\dots(n-(k-2))(n-(k-1))}$$

$$\geq \frac{(i+\frac{k}{4})^{\frac{3k}{4}+1}}{n(n-1)(n-2)\dots(n-(k-2))(n-(k-1))}$$

$$\geq \frac{\left(\left(\frac{n(n-1)(n-2)\dots(n-(k-2))(n-(k-1))}{p}\right)^{\frac{4}{3k+4}} - \frac{k}{4} + \frac{k}{4}\right)^{\frac{3k}{4}+1}}{n(n-1)(n-2)\dots(n-(k-2))(n-(k-1))}$$

$$= \frac{1}{p}$$

Theorem 1 states that, given a desired probability $\frac{1}{p}$ and *any* targeted $k$-combination of a set $S$, we can always find $i$ to obtain a $(k + i)$-combination of the set $S$ that contains the targeted $k$-combination of the same set $S$ with probability at least $\frac{1}{p}$. Hence, among all such $\binom{n}{k+i}$ combinations, at least $\frac{1}{p}$ of them contain the targeted $k$-combination. Because this is true for *any* targeted $k$-combination of a set $S$, each element in $S$ appears in at least $\frac{1}{p}$ of all such $\binom{n}{k+i}$ combinations. We therefore have the following proposition.

**Proposition 1.** *Let $\frac{1}{p}$ be the probability in Theorem 1 that a $(k + i)$-combination of the set $S$ contains the targeted $k$-combination of the same set $S$. Each element in $S$ is in at least $\frac{1}{p}\binom{n}{k+i}$ combinations of a total of $\binom{n}{k+i}$ combinations.*

---

**Algorithm 1** To select $k$ items at random from a set of $n$ items.

1:  **procedure** S($L,n,k$)                                        ▷ a list $L$ of $n$ items and a number $k$
2:      Set $t \leftarrow 0$.                                       ▷ $t$ the total number of of items dealt with so far
3:      Set $m \leftarrow 0$.                                       ▷ $m$ the number of items selected so far
4:      Generate a random number $U$ uniformly at random between 0 and 1.
5:      **if** $(n - t)U \geq k - m$ **then** goto step 8.
6:      Select the next item in $L$ for the sample and increase $m$ and $t$ by 1.
7:      If $m < k$, go to step 4, else the procedure terminates.
8:      Skip the next item in $L$, increase $t$ by 1, and go back to step 4.

---

Moreover, the value $i \geq \left(\frac{n(n-1)(n-2)...(n-(k-2))(n-(k-1))}{p}\right)^{\frac{4}{3k+4}} - \frac{k}{4}$ is tight in a sense that it gives exactly $\frac{1}{p}$ when substituted in the lower bound of $\frac{\binom{n-k}{i}}{\binom{n}{k+i}}$. In other words, if we want to improve this value $i$, we need to improve the lower bound.

Given an appropriate value of $i$ from Theorem 1, we will compute a random set of $k + i$ out of $n$ items. A randomized algorithm that is used to generate a $k$-combination of a set $S$ uniformly at random is due to *Fan, Muller & Rezucha (1962)* and *Jones (1962)* and was described in greater detail in *Knuth (1998)*. For a historical reason, we use the original name called *procedure S*. We describe it as follows. Assume $L = \{1, 2, 3, \ldots, n - 1, n\}$ and $0 < k \leq n$.

By Theorem 1, if we set $i \geq \left(\frac{n(n-1)(n-2)...(n-(k-2))(n-(k-1))}{p}\right)^{\frac{4}{3k+4}} - \frac{k}{4}$ and execute procedure $S(L, n, k + i)$ one time, the chance of obtaining a $(k + i)$-combination of the set $S$ containing the targeted $k$-combination of the same set $S$ is at least $\frac{1}{p}$. By executing procedure $S(L, n, k + i)$ $t$ times independently, the chance of *not* obtaining a $(k + i)$-combination of the set $S$ that contains the targeted $k$-combination of the same set $S$ becomes at most $(1 - \frac{1}{p})^t$.

We remark here that the time complexity of the procedure $S(L, n, k)$ is $O(n)$ and the algorithm will terminate before considering the last record exactly $1 - \frac{k}{n}$ of the time since the last item is selected with probability $\frac{k}{n}$ (*Knuth, 1998*). In other words, most of the time, this algorithm does *not* consider every item in $L$, making it computationally fast. We use $\lceil i \rceil$ when $i$ is not an integer. We note that, in some cases, $i$ may be greater than $n$. For instance, $n = 10,000$ and $k = 20$. In such cases, we will handle it differently. Later we discuss limitations and mitigation for the cases when $i$ is too large.

## CONTENT OPTIMIZATION IN ONLINE ADVERTISEMENT

In this section we discuss how the probability boosting technique can be applied in the online advertising business. In online social media platforms such as Facebook customers can pay to have their advertisements displayed on the platforms. The rates of services may vary according to how often an advertisement must appear on the targeted social media: the higher the frequency, the more expensive the rate. At the same time, these advertisements cannot stay on display all the time in the same sequence of advertisements

---

**Algorithm 2  To display the advertisements with two rates.**

1:   Set $B \leftarrow \varnothing$.

2:   Set $p \leftarrow 2$.

3:   Set $i \leftarrow \left\lceil \left( \frac{n(n-1)(n-2)...(n-(k-2))(n-(k-1))}{p} \right)^{\frac{4}{3k+4}} - \frac{k}{4} \right\rceil$.

4:   Invoke the procedure $S(L, n, k+i)$ to obtain a random $(k+i)$-set.

5:   If all $k$ targeted advertisements are in the resulting $(k+i)$-set then

6:       Add the $(k+i)$-set to $B$.

7:       Display all advertisements in $B$.

8:       Set $B \leftarrow \varnothing$.

9:   Go to step 4.

---

because people may get bored. For the purpose of illustration, suppose an online social media company is paid to display 5,000 advertisements on its platform. Among these 5,000 advertisements, two of them are very expensive and must be displayed at least 50% of the time according to the contracts the company has made with customers. Algorithm 2 that meets this particular requirement can be defined as above. In this specific example, $L$ is a list of 5,000 advertisements with $n = 5{,}000$, $k = 2$, and $i = 690$. By the virtue of Theorem 1, it is with probability at least $\frac{1}{2}$ that the two very expensive advertisements are displayed on the company's social media platform with the other $i$ advertisements on step 6. We assume that this algorithm does not terminate because the advertisements run 24 hours a day everyday.

In another scenario, if there is a single rate for all advertisements and each of the $k+i$ advertisements is to be displayed each time simultaneously with probability at least $\frac{1}{2}$ at $k+i$ different social media outlets, the following algorithm can be used.

Again we assume that the advertisements are displayed without stopping. The correctness of this algorithm follows Proposition 1.

## SERVER SELECTION IN A HETEROGENEOUS ENVIRONMENT

In this section we discuss how to apply the probability boosting technique in Theorem 1 to manage a network of $n$ heterogeneous computer servers. Among these $n$ servers, $k$ of them are very expensive supercomputers with the power of doing $2 \times 10^{18}$ calculations per second (*Oak Ridge National Laboratory, 2022*). Because they are very expensive supercomputers, we can only afford to have a small number of them. The other servers are regular servers with the power of doing $3 \times 10^9$ calculations per second (*phoenixNAP— Global IT Services, 2022*). In a real computer network, in which each computer may be in a different country, we do not see computers physically. All we know are their IP addresses or server identification numbers, which change dynamically. In order to know whether a computer server is regular or not, a test must be performed and this test is costly in terms of communication time. We try to avoid testing *all* $n$ computer servers at all costs due to the potential to cause overwhelming network traffic.

---

**Algorithm 3** To display the advertisements with one rate.

1:     Set $B \leftarrow \varnothing$.

2:     Set $p \leftarrow 2$.

3:     Set $i \leftarrow \left\lceil \left( \frac{n(n-1)(n-2)...(n-(k-2))(n-(k-1))}{p} \right)^{\frac{4}{3k+4}} - \frac{k}{4} \right\rceil$.

4:     Invoke the procedure $S(L, n, k+i)$ to obtain a random $(k+i)$-set.

5:     Add the resulting $(k+i)$-set to $B$.

6:     Display each advertisement in $B$ at each social media outlet simultaneously.

7:     Set $B \leftarrow \varnothing$.

8:     Go to step 4.

---

Our objective is to leverage the $k$ very expensive supercomputers to reduce the overall computational time and at the same time reduce the number of tests to be performed. Let $L = \{1, 2, 3, \ldots, n-1, n\}$ be server identification numbers and $0 < k \leq \lceil f\log_{10}(n) \rceil$, where $0 < f \leq 1$ is a constant. To choose a set of $k$ servers for processing, we use Algorithm 4 that is defined as follows:

The time complexity of Algorithm 4 is $O(nt)$ and the space complexity for $B$ is $O((k+i)t)$. The time to verify the correct $(k+i)$-set is $O((k+i)t)$ because the verification must be done for every $(k+i)$-set to find the $k$ supercomputers, assuming the time taken to verify the types of servers is a very large constant $O(1)$. Additionally, in step 4, Algorithm 1 terminates before considering the last record exactly $1 - \frac{k+i}{n}$ of the time since the last item is selected with probability $\frac{k+i}{n}$ (Knuth, 1998). Therefore, most of the time, Algorithm 4 does not consider every server in $L$ in step 4 so that it is relatively fast. Lemma 1 gives the probability that Algorithm 4 obtains the $k$ supercomputers. The main idea of the proof is similar to tossing a fair coin that has the probability of $\frac{1}{2}$ of obtaining a head and the other $\frac{1}{2}$ of obtaining a tail. To increase the probability of obtaining a head, we can repeat the tosses many times to make sure that at least one of the outcomes is a head.

**Lemma 1.** *Algorithm 4 gives a $(k+i)$-set of servers that contains the $k$ supercomputers with a probability of $1 - \frac{1}{2^t}$.*

*Proof.* By Theorem 1, the procedure $S(L, n, k+i)$ on line 4 returns a $(k+i)$-set that contains the $k$ supercomputers with probability at least $\frac{1}{2}$. Therefore, the probability of *not* getting such a $(k+i)$-set is less than $\frac{1}{2}$ for one execution of the procedure $S(L, n, k+i)$. Because the algorithm repeats the call to the procedure $S(L, n, k+i)$ $t$ times and each call is made independently, the probability that at least one of the $t$ $(k+i)$-sets of servers contains the $k$ supercomputers is at least $1 - \frac{1}{2^t}$. Hence, Algorithm 4 gives a $(k+i)$-set of servers that contains the $k$ supercomputers with probability $1 - \frac{1}{2^t}$.

Observe that if $t = \lfloor \log_2(n) \rfloor$, Algorithm 4 gives a $(k+i)$-set of servers that contains the $k$ supercomputers with a probability of $1 - \frac{1}{n}$, which is with high probability. However, the space complexity becomes $O((k+i)\log_2(n))$ and this fact inevitably requires the number of tests to be the same number as the space complexity. If $i$ is proportional to $n$, this number of tests would be worse than $n - k$ tests in the straightforward deterministic

---

**Algorithm 4** To boost the probability of obtaining the $k$ supercomputers.

1:    Set $B \leftarrow \varnothing$.

2:    Set $p \leftarrow 2$.

3:    Set $i \leftarrow \left\lceil \left( \frac{n(n-1)(n-2)...(n-(k-2))(n-(k-1))}{p} \right)^{\frac{4}{3k+4}} - \frac{k}{4} \right\rceil$.

4:    Invoke the procedure $S(L, n, k + i)$ to obtain a random $(k + i)$-set.

5:    Add the resulting $(k + i)$-set to $B$.

6:    Repeat steps 4 and 5 $t - 1$ more times.

7:    Return $B$.

---

algorithm. Hence, we would like $t$ to be as small as possible to gain a real advantage. The following proof shows that if $n - k > k + i$ and $t < \frac{n-k}{k+i}$ in the execution of Algorithm 4, the number of required tests to verify the $k$ supercomputers in the worst case is strictly less than $n - k$. The idea of the proof is simply noticing that there is a total of $t(k + i)$ servers to be tested and then using the two given conditions to show that the theorem holds.

**Theorem 2.** *If $n - k > k + i$ and $t < \frac{n-k}{k+i}$ in the execution of Algorithm 4, the number of required tests to verify the k supercomputers in the worst case is strictly less than $n - k$ of the straightforward deterministic algorithm.*

*Proof.* Suppose $n - k > k + i$ and $t < \frac{n-k}{k+i}$. The number of required tests to verify the $k$ supercomputers in the worst case depends on the $t$ $(k + i)$-sets of servers from the sampling in Algorithm 4. There is a total of $t(k + i)$ servers to be tested. If $n - k > k + i$ and $t < \frac{n-k}{k+i}$, then $t(k + i) < n - k$. The theorem holds.

To verify that there is a set of numbers for $t$, $n$, $k$, and $i$ such that Theorem 2 holds. We give the following example. Let $n = 10,000$, $k = 2$, and $i = 1,201$. By the virtue of Theorem 1, it is with probability at least $\frac{1}{2}$ that a $(k + i)$-set contains the $k$ supercomputers with probability at least $\frac{1}{2}$. In this case, $k + i = 1,203$, $n - k = 9,998$, and $t = 8 < \frac{9,998}{1,203} = 8.31$. Therefore, the number 9,624 of required tests is strictly less than 9,998, saving 374 tests and this saving also comes with the probability of $1 - \frac{1}{2^8} = 1 - \frac{1}{256}$ of obtaining the $k$ supercomputers by Lemma 1. Hence, this method saves 374 tests of required tests with the probability of 0.99 that the $k$ servers are supercomputers.

In a real situation we could apply this probability boosting technique to select a set of $k$ supercomputers to process a number of jobs in parallel. Suppose there are $j$ jobs to complete using the network of $n$ heterogeneous computer servers. These $j$ jobs come in one at a time. Among these $n$ computer servers, $k = \lceil f\log_{10}(n) \rceil$ of them are very expensive supercomputers with the power of doing $2 \times 10^{18}$ calculations per second. The other $n - k$ servers are regular servers with the power of doing $3 \times 10^9$ calculations per second. Each of these $j$ jobs requires an identical number of calculations $c$ and that each job is divided into $k$ pieces with $\frac{c}{k}$ calculations each. If $k$ regular servers process one of these jobs, it takes $\frac{c}{k \times 3 \times 10^9}$ seconds to complete a job. On the other hand, if $k$ supercomputers process one of these jobs, it takes $\frac{c}{k \times 2 \times 10^{18}}$ seconds to complete the same job. In a parallel processing environment, a slowest server determines the total processing time to complete a job.

---

**Algorithm 5**  **To process the $j$ jobs using $k$ randomly-selected computers.**

1: Obtain a set $B$ of $k + i$ computers by executing Algorithm 4.

2: Obtain a set $F$ of $k$ fastest computers among the $k + i$ computers in $B$.

3: For each incoming job, execute it in parallel on the $k$ computers in $F$.

---

Hence, if we randomly pick a $k$-set from $n$ computer servers, the slowest of these $k$ servers determines the total processing time for the job.

Observe that if Algorithm 4 with an appropriate value of $t$ does not return the set of $k$ supercomputers, the $j$ jobs are still executed by a possibly mixed set of regular and supercomputer servers. No harm is done except that we do not obtain the speedup that we would like to have from the set of $k$ supercomputers. Theorem 3 shows the time expected to complete all $j$ jobs using Algorithm 5. The proof simply uses the fact that we know the probabilities of the case where all $k$ supercomputers are obtained and the case where all $k$ supercomputers are not obtained by the random process in the algorithm and the numbers of calculations in each case.

**Theorem 3.** *If we use Algorithm 5, the expected time to complete all $j$ jobs is $j \times ((1 - \frac{1}{2^t})(\frac{c}{k \times 2 \times 10^{18}}) + (\frac{1}{2^t})(\frac{c}{k \times 3 \times 10^9}))$, where $t$ is the number of loops in Algorithm 4.*

*Proof.* Let $C_i$ be a random variable for the number of calculations job $i$ takes to complete itself and $C = \sum_{i=1}^{j} C_i$ be a random variable for the number of calculations to complete all $j$ jobs.

$$C_i = \begin{cases} \frac{c}{k \times 2 \times 10^{18}} & \text{if } i \text{ is processed by k supercomputers,} \\ \frac{c}{k \times 3 \times 10^9} & \text{otherwise} \end{cases}$$

The expected number of calculations for each job $E[C_i] = \left(1 - \frac{1}{2^t}\right)\left(\frac{c}{k \times 2 \times 10^{18}}\right) + \left(\frac{1}{2^t}\right)\left(\frac{c}{k \times 3 \times 10^9}\right)$ and the expected number of calculations for all $j$ jobs is as follows.

$$E[C] = E\left[\sum_{i=1}^{j} C_i\right]$$

$$= \sum_{i=1}^{j} E[C_i] = j \times \left(\left(1 - \frac{1}{2^t}\right)\left(\frac{c}{k \times 2 \times 10^{18}}\right) + \left(\frac{1}{2^t}\right)\left(\frac{c}{k \times 3 \times 10^9}\right)\right)$$

Hence, the theorem holds.

A few remarks are in order. First, the time complexity of Algorithm 5 is the time complexity of Algorithm 4 in Step 1 and the time complexity for verifying the $k$ servers in Step 2 and the average time to process all $j$ jobs in Step 3. Steps 1 and 2 are executed only one time and the time is $O(nt) + O((k + i)t)$. The average time in Step 3 is the time in Theorem 3. Second, this result depends on the value $t$. As $t$ grows, the expected number of calculations in Step 3 approaches that of supercomputers multiplied by $j$. Third, if we just

pick a set of $k$ computers at random without the probability boosting technique for each job, the expected number of calculations would be as follows.

$$C_i = \begin{cases} \frac{c}{k \times 2 \times 10^{18}} & \text{if } i \text{ is processed by k supercomputers,} \\ \frac{c}{k \times 3 \times 10^9} & \text{otherwise} \end{cases}$$

The expected number of calculations for each job

$$E[C_i] = \frac{1}{\binom{n}{k}}\left(\frac{c}{k \times 2 \times 10^{18}}\right) + \left(1 - \frac{1}{\binom{n}{k}}\right)\left(\frac{c}{k \times 3 \times 10^9}\right) \geq \frac{1}{\left(\frac{en}{k}\right)^k} \times \left(\frac{c}{k \times 2 \times 10^{18}}\right) + \left(1 - \frac{1}{\left(\frac{n}{k}\right)^k}\right)$$

$\left(\frac{c}{k \times 3 \times 10^9}\right)$ because $\left(\frac{n}{k}\right)^k \leq \binom{n}{k} \leq \left(\frac{en}{k}\right)^k$ according to *Cormen et al. (2001)* and the expected number of calculations for all $j$ jobs is therefore as follows.

$$E[C] = E\left[\sum_{i=1}^{j} C_i\right]$$
$$= \sum_{i=1}^{j} E[C_i] \geq j \times \left(\frac{1}{\left(\frac{en}{k}\right)^k}\left(\frac{c}{k \times 2 \times 10^{18}}\right) + \left(1 - \frac{1}{\left(\frac{n}{k}\right)^k}\right)\left(\frac{c}{k \times 3 \times 10^9}\right)\right)$$

In this latter case, the expected number of calculations for all $j$ jobs is much greater than that with the probability boosting technique because $\frac{1}{\left(\frac{en}{k}\right)^k}$ is a lot smaller than $1 - \frac{1}{\left(\frac{n}{k}\right)^k}$. We illustrate this winning advantage with a concrete example. Let $n = 10{,}000$, $k = 2$, $i = 1{,}201$, $t = 8$, and we consider only one job. With the probability boosting technique, the expected time to complete the job is $(1 - \frac{1}{2^8})(\frac{c}{2 \times 2 \times 10^{18}}) + (\frac{1}{2^8})(\frac{c}{2 \times 3 \times 10^9})$. On the other hand, without the probability boosting technique, the expected time to complete the same job is $\frac{4}{e^2(10{,}000)^2}(\frac{c}{2 \times 2 \times 10^{18}}) + (1 - \frac{4}{10{,}000^2})(\frac{c}{2 \times 3 \times 10^9})$. Clearly, with the probability boosting technique, the processing speed is much closer to that of supercomputers while, without the probability boosting technique, the processing speed is much closer to that of regular computers.

## STRING COMPARISON

In biology a protein is a sequence of certain English characters and each position in the sequence has a certain representation. Significant similarity in proteins such as DNA and RNA is a strong evidence that two protein sequences are related by evolutionary changes from a common ancestral sequence. Identifying differences in these protein sequences gives us useful information about a common ancestral relation of proteins and therefore animals. In this section we discuss how the probability boosting technique can be applied to detect differences in protein sequences or character strings in general.

For the sole purpose of illustration, two strings of even length $n$ are exactly the same if and only if a character in each position of the two strings is the same. If there are at least $\lfloor \frac{k}{2} \rfloor$ differences in each half of the strings, the two strings are considered *non-related*. Given two non-related strings of even length $n$, we would like to identify $k$ different positions between the two strings.

---

**Algorithm 6** To detect k different positions in $S$ and $S'$ of length $n$.

1:     Set $p \leftarrow \sqrt{2}$.

2:     Set $k' = \lfloor \frac{\log_{10}(n)}{2} \rfloor$

3:     Set $i_1 \leftarrow \left\lceil \left( \frac{\frac{n}{2}(\frac{n}{2}-1)(\frac{n}{2}-2)...(\frac{n}{2}-(k'-2))(\frac{n}{2}-(k'-1))}{p} \right)^{\frac{4}{3k'+4}} - \frac{k'}{4} \right\rceil$.

4:     Set $i_2 \leftarrow \left\lceil \left( \frac{\frac{n}{2}(\frac{n}{2}-1)(\frac{n}{2}-2)...(\frac{n}{2}-(k'-2))(\frac{n}{2}-(k'-1))}{p} \right)^{\frac{4}{3k'+4}} - \frac{k'}{4} \right\rceil$.

5:     Repeat

6:          Invoke the procedure $S(L_1, \frac{n}{2}, k' + i_1)$ to obtain a random $(k' + i_1)$-set.

7:          Invoke the procedure $S(L_2, \frac{n}{2}, k' + i_2)$ to obtain a random $(k' + i_2)$-set.

8:          Check positions in the $(k' + i_1)$-set in $S_1$ and $S_1'$ for differences.

9:          Check positions in the $(k' + i_2)$-set in $S_2$ and $S_2'$ for differences.

10:   Until there are at least $k'$ differences in both pairs.

11:   Report the $k'$ positions in each half.

A straight forward method is to compare each character of the two strings from position 1 to position $n$. This would take $n - k$ time in the worst case and is not the best way when $k$ is small or $k = \lceil \log_{10}(n) \rceil$. If we, however, pick $k$ positions at random and compare them, the chance that we have correct $k$ different positions is only 1 in $\binom{n}{k}$. Instead, we use Algorithm 6 with the probability boosting technique.

Let $S$ and $S'$ be two given non-related strings of even length $n$ and $S_1$, $S_2$ be the first half and second half of $S$ and $S_1'$, $S_2'$ be the first half and second half of $S'$, respectively. Let $L_1 = \{1, 2, 3, \ldots, \frac{n}{2}\}$ and $L_2 = \{\frac{n}{2} + 1, \frac{n}{2} + 2, \frac{n}{2} + 3, \ldots, \frac{n}{2} + \frac{n}{2} - 1, \frac{n}{2} + \frac{n}{2}, \}$.

Observe that Algorithm 6 is Las Vegas and is always correct when it reports the $k'$ positions in each half. This algorithm always terminates because the two given input strings are non-related. However, the time taken might vary between runs, even with the same input. Hence, in this example, we will bound the running time. Lemma 2 shows the number of loops that can be expected in Algorithm 6 and its variance. The idea of the proof is to observe that the number of loops has a geometric distribution.

**Lemma 2.** *The expected number of loops in Algorithm 6 is at most 2 and the variance is at most 2.*

*Proof.* We need to bound the probability that both halves of the strings have at least $k'$ differences because this is when the algorithm terminates. This probability equals the probability that the first half has at least $k'$ differences and the second half has at least $k'$ differences. By Theorem 1, the probability that each half has at least $k'$ differences is at least $\frac{1}{\sqrt{2}}$. Because positions in each half are independently chosen, the probability that the first half has at least $k'$ differences and the second half has at least $k'$ differences is therefore at least $\frac{1}{2}$. Because this experiment has a geometric distribution, the expected number of loops until there are at least $k'$ differences in both halves is at most 2 and the variance is at most 2.

To see how running time of Algorithm 6 deviates from the expected number of loops, let $X$ be a random variable representing the number of loops in Algorithm 6 and $E(X) = 2$

and $Var(X) = 2$ by Lemma 2. By Chebychev's inequality, we have that $Pr(|X - E(X)| \geq 3) \leq \frac{2}{3^2} = \frac{2}{9}$, which is quite small. We further remark that a clear advantage of this algorithm is the number of times that positions are checked to find differences between the two strings. We give a concrete example when $n = 20,000$, $k = 5$, $k' = 2$, $i_1 = i_2 = 1,380$. In this case the number of positional checks that can be saved on average equals the length of the string-the average number of iterations $\times 2 \times 1,382 = 20,000 - 2.2(1,382) = 14,472$. In the deterministic case all 19,995 positions must be checked in the worst case.

## LIMITATIONS

In Theorem 1, we compute $i$ according to the following inequality.

$$i \geq \left( \frac{n(n-1)(n-2)\ldots(n-(k-2))(n-(k-1))}{p} \right)^{\frac{4}{3k+4}} - \frac{k}{4}$$

In a rare case, $i$ could result in a value greater than or equal to $n - k$, this method simply picks the complete set. This could happen when $n$ or $k$ are large. In most cases, this probability boosting method produces a good value for $i$. We mention here that $n$ does not have to be large in order to use this method because the chance of picking the targeted $k$-set out of $\binom{n}{k}$ is still very small. As the problem of content optimization in online advertisement in the previous section illustrated, one cannot simply do it manually when $n$ is as small as 5,000. Nonetheless, we suggest the following remedies in the case where $i \geq n - k$ whenever they are applicable.

1. Increase the value of $p$ in the computation of $i$. This makes $i$ smaller but may have to trade it with more running time of the algorithm to obtain the desired result.
2. Reduce the value of $n$ and $k$ using the divide and conquer technique illustrated in the example of string comparison. In this case, the more subproblems we have, the smaller the values of $n$ and $i$ become. The divide and conquer technique can also accommodate parallel computation to speed up the overall process.
3. Because $\binom{n}{k} = \binom{n}{n-k}$, some applications may allow us to pick $k$ or $n - k$, whichever is smaller.

## CONCLUSION

In this article we considered a type of combinatorial optimization that involves choosing a targeted $k$-subset of $n$ objects. We discussed the motivation and the probability-boosting technique that can be used to increase the probability of choosing $k$ out of $n$ objects. Three different applications of the probability-boosting technique were then illustrated. We summarize the advantages of each example in Table 1.

In the end potential limitations of the probability-boosting technique were discussed and mitigation was suggested. We are convinced that the three examples in this article are among many applications of this probability-boosting technique to be discovered. In terms of future research, we may investigate the application of the probability-boosting

technique in a combinatorial optimization problem whose $k$ items are not obvious and/or come with some constraints such as the hybrid flow-shop scheduling problem (*Deng et al., 2024*).

### Funding
The author received no funding for this work.

### Competing Interests
The author declares that he has no competing interests.

### Author Contributions
- Sanpawat Kantabutra conceived and designed the experiments, prepared figures and/or tables, authored or reviewed drafts of the article, and approved the final draft.

### Data Availability
This is a theoretical article.

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
