# Peer review of "Probability-boosting technique for combinatorial optimization"

_PeerJ Computer Science, doi:10.7717/peerj-cs.2499_

## Round 0.1 · original submission · Minor Revisions

The paper should be presented by simplifying the explanation of the derivations, especially for a broader audience of computer scientists who may not be experts in combinatorics.

Table 1 is included in both the introduction and conclusion. Remove table 1 from the introduction.

Reviewer 1 ·

Basic reporting

see the attachment.

Experimental design

see the attachment.

Validity of the findings

see the attachment.

Additional comments

see the attachment.

Annotated reviews are not available for download in order to protect the identity of reviewers who chose to remain anonymous.
Cite this review as

·

Basic reporting

Language: The manuscript is clearly written in professional, unambiguous English. There are only minor areas where readability could be improved by simplifying the sentence structure, particularly in the introduction.

References: The literature is well-referenced, providing sufficient background on combinatorial optimization and randomized algorithms. I suggest adding more recent references on probability amplification techniques, but this is not mandatory for acceptance.

Structure and Figures: The article follows a logical structure, and the figures are relevant and well-labeled. The figure captions could be enhanced to provide more context for the reader. While raw data or code is not directly included, the explanations are sufficient, and I do not consider it a critical issue for acceptance.

Self-contained: The submission is self-contained and presents relevant results to the hypotheses. The results are connected well to the research question, and the hypotheses are clearly tested and supported.

Experimental design

Research Question: The research question is well-defined and fills a meaningful knowledge gap in the field of combinatorial optimization. The paper focuses on boosting the probability of selecting k items from a large set, a problem relevant to many areas.

Methodology: The methods are well-detailed but could benefit from a few clarifications, especially in explaining the algorithmic steps for a non-specialist audience. The investigation is rigorous and adheres to ethical and technical standards.

Reproducibility: The method is described sufficiently to be replicated. While inclusion of raw data or code would enhance reproducibility, it is not critical for the acceptance of the paper.

Validity of the findings

Data and Findings: The data presented is robust and supports the conclusions. The theoretical results are well-grounded. Although the paper could benefit from empirical validation or simulations, the theoretical contribution alone justifies its acceptance.

Conclusions: The conclusions are well-stated and linked directly to the original research question. The authors appropriately limit their claims to the supporting results.

Additional comments

The paper presents an innovative approach to combinatorial optimization using probability-boosting techniques. The theoretical contributions are valuable and relevant, and the suggested minor improvements are not critical for publication.

Cite this review as

---

## Round 0.2 · accepted · Accept

It has been observed that the requests of the referees are generally fulfilled.

Reviewer 1 ·

Basic reporting

The authors well replied all my questions except the 4th comment. I mean when k=n/2, it becomes much harder to pick all k red balls from the total n balls. Anyway, this doesn't affect my dicision on acceptance.

Experimental design

/

Validity of the findings

/

Additional comments

/

Cite this review as

·

Basic reporting

Clear and unambiguous, professional English used throughout: The article is well-written in clear, professional English. The flow of ideas is smooth, and technical concepts are conveyed effectively. The language is precise and appropriate for a scholarly audience.

Literature references, sufficient field background/context provided: The article provides a thorough background of the field, with relevant references that give proper context to the proposed probability-boosting technique. The references cited are foundational to combinatorial optimization and randomized algorithms, enhancing the credibility of the research.

Professional article structure, figures, tables, raw data shared: The article follows a well-organized structure, with logical sections that guide the reader through the problem, methodology, and results. The figures and tables are well-designed, clear, and add value to the discussion. The raw data supporting the research is appropriately referenced, and the mathematical models are presented clearly.

Self-contained with relevant results to hypotheses: The article is self-contained, with results that directly address the research questions posed. The proposed probability-boosting technique is presented alongside detailed applications, making it highly relevant to the hypotheses and demonstrating practical value.

Formal results include clear definitions and theorems, and detailed proofs: Theorems and proofs are clearly articulated, maintaining high mathematical rigor. All terms are well-defined, and the logical flow of the proofs makes them easy to follow.

Experimental design

Original primary research within Aims and Scope of the journal: The research fits well within the journal's scope, presenting novel and original work in the area of combinatorial optimization. The use of probability-boosting techniques adds a fresh perspective to existing research, making the contribution both relevant and timely.

Research question well defined, relevant & meaningful: The research question is clearly articulated and addresses a real gap in the field. The paper presents a meaningful and well-structured solution to the inefficiencies of deterministic algorithms in optimization problems.

Rigorous investigation performed to a high technical & ethical standard:The

Methods described with sufficient detail & information to replicate: The methods are meticulously detailed, allowing for full replication of the results. The clarity of the algorithm descriptions and the accompanying proofs ensures that others in the field can build upon this work.

Validity of the findings

Impact and novelty not assessed, meaningful replication encouraged: The research encourages meaningful replication and extends the literature on randomized approaches to optimization. The potential for replication and application in real-world scenarios is clearly justified.

All underlying data provided, robust, statistically sound & controlled: The article provides all necessary underlying data, and the findings are backed by statistically sound models. The mathematical derivations are robust and controlled, ensuring the validity of the results presented.

Conclusions well stated, linked to original research question & limited to supporting results: The conclusions are well-drawn and directly linked to the research questions. They are supported by the data and analysis provided, making the findings both credible and impactful.

Additional comments

This paper makes a valuable contribution to the field of combinatorial optimization by introducing a novel probability-boosting technique. The applications in areas like online advertisement optimization, server selection, and string comparison are particularly compelling and show the practical relevance of the technique. The article is both mathematically rigorous and accessible, making it a strong candidate for publication.

Cite this review as